# Influence of Fire-Induced Heat and Moisture Release on Pyro-Convective Cloud Dynamics during the Australian New Year's Event: A Study Using Convection-Resolving Simulations and Satellite Data

Lisa Janina Muth<sup>1</sup>, Sascha Bierbauer<sup>1</sup>, Corinna Hoose<sup>1</sup>, Bernhard Vogel<sup>1</sup>, Heike Vogel<sup>1</sup>, and Gholam Ali Hoshyaripour<sup>1</sup>

<sup>1</sup>Institute of Meteorology and Climate Research Troposphere Research, Karlsruhe Institute of Technology (KIT)

Correspondence: Lisa Janina Muth (lisa.muth@kit.edu)

#### Abstract.

Understanding pyro-convective clouds is essential. These clouds transport significant quantities of aerosols and gases into the upper atmosphere, and therefore influence atmospheric composition, weather, and climate on a global scale. This study investigates the dynamics of pyro-convective clouds during the Australian New Years Event 2019/2020 using convection-resolving simulations that incorporate the effects of sensible heat and moisture released by fires. These effects are modeled through parameterizations using retrievals from the Global Fire Assimilation System (GFAS). The results show that the plume top height remains unchanged when accounting for fire-induced heat and moisture release in regions where convective cells form independently of the fire. In areas with the most intense fires, the sensible heat and moisture release from the fire provide the necessary buoyancy for enabling the formation of pyro-convective clouds. Pyro-convective clouds lift aerosol masses up to altitudes of 12.0 km. During their formation, the cloud top height more than doubles compared to a reference simulation in which such clouds do not develop. Additionally, the plume height increased is on average just 0.87 km by fire-induced heat and moisture in cloud-free areas. We demonstrated that sensible heat release is the primary contributor to pyro-convective cloud formation. However, the release of moisture enhances the formation process and increases the lifetime of the pyro-convective cloud. Comparisons with observational data show that the plume's distribution and height are underestimated. However, the simulations align well with observations after a 5-6 hour delay, indicating that pyro-convective cells are accurately modeled but occur later than observed.

#### 1 Introduction

Pyrocumulonimbus (pyroCb) clouds emerge from intense wildfires, generating lightning, hail, downdrafts, and tornadoes, despite minimal precipitation (Fromm et al., 2022). Smoke particles and gases get transported by pyroCb events up to the upper troposphere and lower stratosphere. This affects global climate by altering atmospheric composition and radiative balance (Fromm et al., 2022). The formation of pyroCbs is highly sensitive to atmospheric stability and fire intensity that can change throughout the day (Luderer et al., 2006).

The general assumption is that while a stable atmosphere typically suppresses vertical air movement, limiting wildfire plume heights and convection in the morning, as the day warms, the atmosphere becomes more unstable. This instability allows for higher plume heights and more vigorous pyro-convection, potentially forming pyroCbs (Luo et al., 2022). Wildfires follow a diurnal cycle as well: cooler morning temperatures and higher humidity reduce fire intensity and spread. As temperatures rise and humidity drops during the day, fire intensity and spread increase, with stronger winds further enhancing fire spread. In the evening, lower temperatures and higher humidity reduce fire intensity (Andela et al., 2015; Loudermilk et al., 2022). This leads to a majority of pyroCb clouds forming and reaching maturity in the late afternoons (Fromm et al., 2010). However, there are exceptions to this typical diurnal cycles of the meteorology and fires. As outlined by Luderer et al. (2006), cold fronts can induce significant temperature drops before sunset, which are substantially greater than the usual diurnal variations experienced in the late afternoon. Furthermore, recent studies have shown an increase in nighttime fire activity, particularly in larger wildfires. This increase is attributed to warmer and drier nighttime conditions, which can sustain fire activity throughout the night (Balch et al., 2022).

Although pyroCb clouds are a common and well-studied phenomenon, much remains unknown about their behavior, energetics, history, and impact on the Earth-atmosphere system (Fromm et al., 2022). This challenges the reliable simulation of pyro-convective clouds. The failure to accurately simulate these clouds is accompanied by an incorrect calculation of plume height as the formation mechanism of pyro-convective clouds releases latent heat which generates additional buoyancy. The effect of fire on meteorological variables and, consequently, on pyro-convective cloud formation is often not included in global and regional models. This omission leads to errors in the injection height of gases and particles, subsequently affecting their transport. To parameterize these processes accurately, a comprehensive understanding of the interplay between fire-induced buoyancy, latent heat release, and atmospheric stability is essential.

Significant progress has been made in understanding these phenomena. Numerous studies with coupled fire-atmosphere models have addressed the uncertainties of fire-atmosphere interactions by accounting for fire dynamics. For example, research by Clark and Packham (1996); Clark et al. (2004) employs fine grid resolutions ranging from 4 meters to 120 meters. These studies primarily focus on wind changes induced by the fire and how these changes impact the fire itself. Despite these advancements, further research is needed to fully comprehend the complexities of pyro-convective transport and its broader implications. Kochanski et al. (2013); Kiefer et al. (2010, 2016, 2018) use fire-atmosphere models nested to in coarser grid resolution to simulate meso-scale effects. However, it remains partially unclear how fires influence cloud and plume dynamics, and consequently long range transport, due to the spatial limitations of the simulation domains. The study by Trentmann et al. (2006) focuses on pyro-convective clouds and explicitly simulates plume rise. It concludes that sensible heat release initiates convection, while latent heat release from condensation and freezing dominates the total energy budget. The study finds that the available moisture is primarily entrained, with negligible contribution from fire-released moisture. Luderer et al. (2006) further investigate these findings through sensitivity studies, concluding that meteorological conditions play a dominant role in pyro-convection. They find that the emission of water vapor is less important for the emission height than sensible heat release but enhances the aerosol amount transported to the tropopause level. Additionally, they find that the dynamics and evolution of pyroCbs are weakly sensitive to aerosols acting as cloud condensation nuclei (CCN). This is in contrast to studies by Reutter

et al. (2014) and Chang et al. (2015), who highlight the significant role of aerosols in influencing pyro-convective cloud dynamics and microphysics. They underscore the importance of including detailed aerosol-cloud interactions in high-resolution atmospheric models to accurately simulate cloud formation and precipitation processes. Lee et al. (2020) further analyze that the aerosol effect on pyroCb development is more significant in cases of weak-intensity fires compared to strong-intensity fires. Findings by Kablick III et al. (2018) highlight the significant impact of surface heat fluxes on pyroCb properties and suggests that pyroCb events could influence lower stratospheric water vapor. All this outlines the complexity and variety of processes that influence pyro-convective cloud formation. Understanding these dynamics is crucial, not only from a cloud microphysical perspective but also for aerosol plume development, as plume height is a key factor in plume transport (Val Martin et al., 2006). Knowledge of the injection height is essential for accurately parameterizing injection heights in global transport models and, therefore, for reliably calculating transport.

The smoke emission and dispersion are influenced by the composition, structure, and condition of the fuel, as well as the weather and topography. Consequently, each fire is unique, making it challenging to accurately capture this variability in simulations. Additionally, the limited in situ measurements and the significant spatial and temporal variability of biomass burning make it challenging to accurately monitor fire characteristics and retrieve necessary inputs for plume resolving simulations. To overcome the dependence on individual measurements, we developed a method to parameterize sensible heat and moisture release in models using satellite retrievals from the Global Fire Assimilation System (GFAS), which are based on MODIS observations (Kaiser et al., 2012; CAMS, 2021). As the name suggests, GFAS provides a global dataset that delivers information about a fire within 24 hours of its occurrence. We test our developments by simulating part of the Australian New Year's Event (ANY) event in a limited area mode and resolved convection. ANY refers to an extreme outbreak of pyroCbs, which occurred in South-East Australia around New Year 2019/2020 (Peterson et al., 2021). There are 38 pyroCbs reported between the 29th of December 2019 and 4th of January 2020, divided into 18 sub-events (Peterson et al., 2021). This pyroCb activity resulted in an emission of approximately 1.0 Tg of biomass burning aerosol into the lower stratosphere and had an impact on the atmospheric dynamics, chemistry and the radiation budget (Peterson et al., 2021).

The extreme fire weather conditions occurred when a cold front passed through the region December 30. Additionally, a negatively tilted upper-tropospheric trough, interactions with topography, the presence of low-level overnight jets, and horizontal boundary layer rolls further exacerbated the situation. In combination, these factors create conditions for rapid fire spread and intense pyroCb activity, especially during the night.

In this study, convection-resolving simulations are performed and analyzed to determine if the atmospheric impact of these intense fires is captured and to answer the following research questions:

- 1. How do fire-induced heat and moisture release affect plume and cloud formation under unstable atmospheric conditions?
- 2. What are the discrepancies between simulated and observed plume top heights and spreads, and how suitable are satellite-constrained fire data from the GFAS for accurately simulating pyro-convective clouds?

In the following section, the model system, the developed parameterizations, and the simulation setup of the performed experiment are described. Then, the results are analyzed and discussed, and the conclusions are presented.

# 2 Methods and Materials

100

110

# 2.1 The ICON-ART modeling system

The model used for this work is the ICOsahedral Nonhydrostatic (ICON) numerical weather and climate model. ICON solves the full three-dimensional non-hydrostatic and compressible Navier-Stokes equations on an icosahedral grid (Zängl et al., 2015). The ICON model is able to perform seamless simulations of various processes from local to global scales (Heinze et al., 2017; Giorgetta et al., 2018).

Furthermore, the ART (Aerosol and Reactive Trace gases) module is enabled. It includes the emission, transport, physicochemical transformation and removal of aerosols and trace gases (Rieger et al., 2015). Detailed descriptions are given in Rieger et al. (2015), ? and Muser et al. (2020).

#### 2.2 Parameterization of heat and moisture release and aerosol emission

A parameterization for sensible heat and moisture release and aerosol emission based on satellite retrievals from GFAS is developed. GFAS uses the FRP from NASA's MOD14 product. MOD14 includes thermal radiation observations ( $\lambda \sim 3.9~\mu$ m-11  $\mu$ m) from the polar-orbiting satellites MODIS Aqua and Terra (Giglio, 2007; Justice et al., 2011). For the parameterization of the sensible heat release the FRP is used. The GFAS data set provides the FRP, and therefore, the radiative fraction of total heat release. The FRP is multiplied by a factor of 10 to retrieve the total energy released by the fire, as it is proposed by Val Martin et al. (2012) and further applied in Ke et al. (2021). A factor of 0.55 is applied to convert the total energy to convective energy. The factor is taken from Freitas et al. (2006), following McCarter and Broido (1965) and is similar to the convective fractions of 0.518 and 0.52 proposed in Freeborn et al. (2008) and Morandini et al. (2013), respectively. The FRP is weighted with the diurnal cycle function proposed by Andela et al. (2015) and applied by Walter et al. (2016) in COSMO-ART to account for peak fire intensity in early afternoons, given in Equation 1 and visualized in Appendix A1.

$$d(t_1) = \omega + \frac{1}{\sigma\sqrt{2\pi}} \exp\left(-\frac{1}{2}\left(\frac{t_1 - t_0}{\sigma}\right)^2\right) \tag{1}$$

Here  $\omega$  is a weighting, which is set according to the vegetation type in the respective grid cell.  $\omega$  is 0.039 for tropical forests, 0.018 for savannas and 0.003 for grassland.  $t_1$  is the local solar time,  $t_0$  is the expected value of maximum emission set to 12.5 and  $\sigma$  is the standard deviation, set to 2.5. The heat release is implemented as sensible heat flux from the surface to the atmosphere. This leads to a sensible heat release by the fire  $sh_{fire}$  of:

$$sh_{fire} = FRP \times 5.5 \times d \tag{2}$$

The FRP and the  $sh_{fire}$  both have the unit W m<sup>-2</sup>.

The implementation of the moisture release includes combustion moisture, with an emission ratio of 0.75 H<sub>2</sub>O/(CO+CO<sub>2</sub>) (Parmar et al., 2008). The CO and CO<sub>2</sub> emission fluxes from GFAS are scaled and emitted in the ICON specific humidity tracer. Additionally, fuel moisture is emitted. The fuel moisture is divided into dead and live components and follows thresholds from

Nolan et al. (2016) and Deb et al. (2020). Assuming 30% dead and 70% live fuel, the approximate fuel moisture is 75.42%, multiplied by the GFAS combustion rate. The live-to-dead fuel ratio is taken from Hines et al. (2010).

$$qv_{fire} = (0.75 \times (m_{CO} + m_{CO_2}) + 0.7542 \times m_{load}) \times d$$
(3)

The moisture emission flux by the fire,  $qv_{fire}$  in kg m<sup>-2</sup> s<sup>-1</sup>, is calculated, according to equation 3, using mass fluxes of CO  $(m_{CO})$  and CO<sub>2</sub>  $(m_{CO_2})$  and the combustion rate  $m_{load}$ . All three have the unit kg m<sup>-2</sup> s<sup>-1</sup>. The moisture emission flux is weighted with a diurnal cycle function d. Next, the moisture emission flux is converted to mass mixing ratio and added to the specific humidity tracer at the lowest model level.

30 For the aerosol emission, the GFAS black carbon  $m_{BC}$  and organic carbon  $m_{OC}$  fluxes are combined, weighted by the diurnal cycle function, as stated in equation 4, and subsequently emitted into the lowermost model level, analog to the moisture emission.

$$e_{fire} = (m_{OC} + m_{OC}) \times d \tag{4}$$

The GFAS daily mean mass fluxes  $m_{OC}$  and  $m_{OC}$  fluxes are provided in kg m<sup>-2</sup> s<sup>-1</sup>, and therefore,  $e_{fire}$  has the unit 35 kg m<sup>-2</sup> s<sup>-1</sup>.

# 2.3 Model Configuration

140

150

For this work, a limited area mode simulation is performed. The area of the domain is shown in Figure 1 in the black box, which is approximately 340 km in length (north-south) and 230 km in width (east-west). Prior to the experimental simulations, a global simulation with a grid spacing of 13 km is conducted to obtain the input data for the boundary conditions. This global simulation is initialized using the German Weather Service (DWD) analysis product and does not account for fire impacts on the meteorological variables. The experimental simulations are also initialized with the DWD analysis product and meteorological variables of the boundary conditions are read in every 30 min. The grid spacing is 0.6 km and there are 125 vertical levels from the surface to a maximum height of 30 km. The level thickness increases from in average 95 m in the lowermost level to 550 m at the top. Due to the high spatial resolution, the schemes for convection, subgrid-scale orographic effects (blocking and gravity wave drag), and non-orographic gravity wave drag are de-activated Dipankar et al. (2015). However, this study does not consider aerosol-cloud and aerosol-radiation interaction. For cloud microscopical processes a single-moment scheme is used that predicts the categories cloud water, rain water, cloud ice and snow. The simulations start on the 29th of December 18:00 UTC, which corresponds to 05:00 Australian Eastern Daylight Time (AEDT) (on December 30) and last for 20 hours. The fire emissions are initialized based on assimilated FRP observations provided by the GFAS. The particle emissions include black and organic carbon that are emitted in the lowermost model layer. The size distribution of the aerosols is approximated by a log-normal distribution with a median number diameter  $d_n = 70$  nm and a standard deviation of  $\sigma = 2.0$ . The simulation does not consider atmospheric chemistry, nucleation or condensation, but coagulation.

Three simulation experiments were performed: a reference experiment termed "REF" that only accounts for aerosol emission and moisture and heat release are neglected. A second experiment termed "SH" where only sensible heat released by the fire is

**Figure 1.** The black box shows the simulation domain and FRP input. The green box makes the area with the largest FRP in approximately the center of the domain.

accounted for, and a third experiment termed "SHLH" where both sensible heat and moisture release are enabled. The FRP in the experiment domain is shown in Figure 1. During the simulation, a peak sensible heat release of 2.24 kW m<sup>-2</sup> is reached. The peak fire-induced water vapor flux reaches  $1.07 \times 10^{-6}$  kg m<sup>-2</sup> s<sup>-1</sup> and assuming the latent heat of condensation for water is approximately 2257 kJ kg<sup>-1</sup>, the a potential additional latent heat flux can reach a maximum of 0.02 W m<sup>-2</sup>.

# 2.4 Height retrievals

The NASA 3D wind retrieval algorithm, as described by Carr et al. (2018, 2019, 2020), is employed to determine the height of plumes and clouds. This algorithm leverages stereo imaging, which utilizes geometric parallax to retrieve feature heights. By integrating data from geostationary (GEO) and low-earth orbit (LEO) satellites, it generates three-dimensional (3D) atmospheric motion vectors (AMVs) through a multi-platform, multi-angle stereoscopic approach. The term "3D Winds" refers to the three-dimensional positioning of horizontal AMVs within the atmosphere. Observing the parallax of a feature from two different vantage points (stereo) provides direct information about its height. For this study, the LEO-GEO retrieval method is utilized. The LEO satellite data comes from Terra and Aqua MODIS Level 1B in the blue band (459–479 nm) with a 500 m resolution. The GEO satellite data is from Himawari-8's blue band (430–480 nm). The Advanced Himawari Imager (AHI), operated by the Japan Meteorological Agency, has a 10-minute temporal resolution that is used to track feature movement. MODIS data is then used to calculate parallax, determining AMVs and height. A quality flag is used to remove poor retrievals.

# 170 2.5 Definitions and analysis methods

In the following chapter, the analysis of plume and cloud heights is conducted. Therefore, the plume is defined as a grid cell that exceeds an aerosol mass mixing ratio of  $5\times10^{-8}$  kg m<sup>-3</sup> and a cloud is defined as a grid cell that exceeds a mass mixing ratio of Liquid Water Content (LWC) + Ice Water Content (IWC) of  $5\times10^{-6}$  kg m<sup>-3</sup>. Respectively, the top height is the level with the highest altitude exceeding this threshold.

Further analysis involves the temporal evolution of the plume top heights for a cloud-free plume, a plume with clouds, and a plume with pyro-convective clouds. For this analysis a threshold of 1 kg m<sup>-3</sup> is chosen for the plume. By adopting a higher threshold, we ensure that the diagnosed plume top reflects the dynamically relevant upper extent of the convective column, rather than transient or diluted features near the top of the plume. The presence of clouds within the plume is defined via a threshold for the sum of LWC and IWC within a grid cell. The plume is considered cloud-free if the grid cell has a mass mixing ratio of LWC+IWC smaller than  $1 \times 10^{-32}$  kg m<sup>-3</sup>. In the plume with clouds case, LWC+IWC within a grid cell must be greater  $1 \times 10^{-32}$  kg m<sup>-3</sup>. The plume with pyro-convective clouds is defined by an increase in aerosol mass mixing ratio by  $1 \times 10^{-6}$  kg/m<sup>3</sup> and an increase of LWC+IWC by  $5 \times 10^{-8}$  kg m<sup>-3</sup> within a grid cell compared to the REF experiment. Since there is no pyro-convective cloud in the REF experiment, the grid cells corresponding to the SHLH experiments are used. The displayed top heights are the mean value of the 100 highest plume top heights is calculated for each experiment over time.

# 3 Results


# 3.1 Comparison of model results and observations

To begin, the simulated plume height is compared to observational data to assess the accuracy of our model. Therefore, the simulated plume top heights are compared to the retrieved top height from the NASA 3D wind algorithm. It should be noted that the resolution of the retrieval is approximately 2.2 km, which is coarser than the simulation with a grid spacing of 600 m. Therefore, the simulation is displayed with a mask that maps the observation points to the closest grid point.

The observations in Figure 2a show aerosols and clouds above the fires with heights ranging between 5 and 12 km. Because the retrieval is not able to separate clouds from aerosols the plots display either the plume or the cloud top height. The SHLH experiment, in comparison to the observations, shows smaller elevated areas of the plume and has lower maximum heights.

The plume in the north-eastern center of the domain remains below 5 km. Close to the fire areas cloud heights above 10 km are simulated, but these clouds form independent of the fire, as these are also simulated in REF (Appendix A2). The time of observation is 5.5 hours into the simulation at 10:30 AEDT. This indicates that particles are only emitted for 5.5 hours and fire intensity and emission flux has not peaked, whereas the observations include the background aerosols of fires burning for days and originating from further away.

Figure 2c-d shows the comparison of cloud and plume height retrieval and the SHLH experiment for December 30, 2019, at 14:30 AEDT. The observations in Figure 2c show plume and cloud heights in the range between 5 and 12 km above the fire

**Figure 2.** Plume and cloud top heights a) retrieved from the NASA 3D wind algorithm, b) simulated for the SHLH experiment for December 30, 2019, at 10:30 AEDT and c) retrieved from the NASA 3D wind algorithm, d) simulated for the SHLH experiment for December 30, 2019, at 14:30 AEDT.

**Figure 3.** The averaged FRP assumed in the simulations (black line), with the standard deviation in gray. The red dots represent the mean MODIS active fire FRP for each hour with measurements during the simulation period, along with the standard deviation.

and in the south-western part of the domain heights above 12 km are reached. In the SHLH experiment 2d, heights above 10 km are simulated, which result from clouds independent of the fire, as outlined in AppendixA2. The simulation time is 9.5 hours and the simulated plumes in the area of the green box are again underestimated in the distribution but selectively match the observed top heights.



To better understand these differences a thorough understanding of the assumptions made and underlying processes is necessary. The fire input data remain a major source of uncertainty. The daily mean values provided by GFAS and the assumed generally applicable diurnal cycle are crude approximations. This is illustrated in Figure 3. Firstly, it is evident that the magnitude of the assumed FRP for the simulation underestimates the FRP observed by MODIS. This is expected, as the assumed FRP for the simulation is based on GFAS, which provides a daily mean value averaged over a  $1^{\circ} \times 1^{\circ}$  grid, compared to the MODIS active fire data at a 1 km resolution. The MODIS data have limited measurements due to the overpass times; however, they do not show a clear diurnal cycle as assumed in the parameterization. On the contrary, the measurements indicate high FRP during the night. As mentioned, for this particular event, high nighttime fire activity and pyroCb formation were reported. This leads to structural limitation in our assumption of fire activity and may contribute to the underestimation of simulated plume height and spread compared to observations, as the simulation starts in the early morning, likely underestimating fire intensity. While this limitation is acknowledged, the aim of this study is not to achieve an optimized case-specific simulation of the ANY event. Rather, we seek to evaluate how well pyro-convective processes can be represented using a standard model

configuration. Therefore, we proceed with the analysis of the experiment. The following section elaborates on the impact of fire-induced heat and moisture release on plume evolution and cloud formation.

# 220 3.2 Plume height and cloud formation


First, the impact of both sensible heat and moisture release (SHLH experiment) on plume height and cloud formation is quantified, before discussing the contribution of each individual effect (SH compared to SHLH).

Figure 4 displays temporal evolution of the plume top height in the REF and SHLH experiments. The first row (a-c) displays

**Figure 4.** Temporal evolution of the plume top height in the REF experiment a) 13:00 AEDT, b) 16:00 AEDT, c) 19:00 AEDT, plume top height in the SHLH experiment d) 13:00 AEDT, e) 16:00 AEDT, and f) 19:00 AEDT. The central part is marked with a green box.

the temporal evolution of the plume top height in the REF experiments. At 13:00 AEDT the plume top height peaks in the north-west corner of the domain with a maximum height of 13.71 km. At 16:00 AEDT, in Figure 4b, 2 more plumes rise above 10 km. The elevated plumes are transported southeast/east in Figure 4c. The experiments SHLH and REF exhibit notable similarities in distribution and maximum heights. The main difference between the experiments is observed in the center of the

domain. This area is marked with a green box, as in Figure 1. In the center of the domain maximum heights up to 12.8 km are reached for the SHLH experiment, which is a significant increase, to the REF experiment with plume heights remaining below 2.5 km.

The area within the green box, with the largest emissions, might suggest that a critical sensible heat and moisture release has to be reached for pyro-convective cloud formation. However, the plume top height resulting from these fires is comparable to those originating from the north-western and southeastern parts of the domain, where convective clouds form independently from the fire.

**Figure 5.** Temporal evolution of the LWP+IWP in the REF experiment a) 13:00 AEDT, b) 16:00 AEDT, c) 19:00 AEDT, difference (SHLH-REF) in LWP+IWP experiment d) 13:00 AEDT, e) 16:00 AEDT, f) 19:00 AEDT. The central part is marked with a green box.

Figure 5 displays temporal evolution of the Liquid and Ice Water Path (LWP and IWP) in the REF experiment a-c. The LWP + IWP, in Figure 5a, indicate cloud formation at the northern and western boundaries of the domain. This suggests that clouds form without considering the fire's effect on meteorological variables. Although, it is possible to identify individual convective cells that overlap with the fire area. The cloud cover moves and spreads from the western boundary to the southeast, resulting

in a nearly diagonal cloud formation throughout the domain by 19:00 AEDT. As discussed in the introduction, the ANY event was characterized by a passing cold front that created atmospheric instability and convective clouds, and this is captured by the simulation. The second row (d-f) displays the difference in LWP + IWP of the SHLH-REF experiment. Figure 5d shows some noise in cloud formation due to the release of sensible heat and moisture. Within the green box the LWP + IWP starts to increase and in Figure 5e cloud formation increases up to 4.46 kg m<sup>-2</sup>. Last, Figure 5f primarily exhibits noise in cloud formation. Overall, the fire-induced heat and moisture lead to additional cloud formation in some regions with pre-existing clouds, which has a negligible influence on the plume top height. The temporal evolution in the region marked with the green box, shows a clear increase in cloud water and ice, which indicates convection explains the increase in plume top height in Figure 4.


**Figure 6.** Temporal evolution of the LWP+IWP in the SH experiment a) 13:00 AEDT, b) 16:00 AEDT, c) 19:00 AEDT, difference (SHLH-SH) in LWP+IWP experiment d) 13:00 AEDT, e) 16:00 AEDT, f) 19:00 AEDT. The central part is marked with a green box.

|      | CIN [J kg <sup>-1</sup> ] | CAPE [J kg <sup>-1</sup> ] | CIN GB [J $kg^{-1}$ ] | CAPE GB[J kg <sup>-1</sup> ] |
|------|---------------------------|----------------------------|-----------------------|------------------------------|
| REF  | 351                       | 644                        | 403                   | 190                          |
| SH   | 228                       | 1307                       | 228                   | 914                          |
| SHLH | 228                       | 1309                       | 227                   | 927                          |

**Table 1.** Surface based CIN (Convective Inhibition) and CAPE (Convective Available Potential Energy) in the fire areas outside of the green box and inside the green box (marked with GB) for the REF, SH, SHLH experiments on December 30, 16:00 AEDT. The values are given J kg<sup>-1</sup>.

In the next step, the impact of moisture release on cloud formation is analyzed. The temporal evolution of LWP + IWP in the SH and the difference in LWP + IWP between the SHLH-SH experiments are shown in Figure 6, structured the same as Figure 5.

The SH shows remarkable similarities to REF, except in the region of the green box. There, a convective cloud forms, with LWP+IWP values  $5.37 \text{ kg m}^{-2}$  at 16:00 AEDT, and dissipates at 19:00 AEDT. This outlines that fire-induced moisture release in not necessary for the formation of this pyro-convective cloud. The impact of moisture release is highlighted in Figure 6d-f. The cloudy regions outside the green box predominantly display noise, with the signal intensity increasing over time. Figure 6d shows a small overall increase, with the total increase in LWP + IWP in that area by 10.6%. The LWP + IWP in Figure 6e decreases by 10.8% and only increase by 1.3% in Figure 6f.





Figures 4-6 outline that the impact of fire-atmosphere interaction is most dominant in the area of the green box. To understand this further, we analyze the atmospheric ability. Table 1 shows how the fire-atmosphere interaction impacts surface-based Convective Inhibition (CIN) and surface-based Convective Available Potential Energy (CAPE) of the green box (GB) and elsewhere for December 30, 16:00 AEDT. The average CIN and CAPE values are calculated within the fire area for the corresponding experiments. The CIN in the REF model indicates that outside of the green box, the atmosphere is less resistant to the initiation of convection. Furthermore, the CAPE values suggest that more energy is available for convection, which can lead to stronger and more vigorous updrafts outside of the green box. This is consistent with the formation of convective clouds at the borders of the domain, but the absence of clouds within the green box. Comparing the CIN of the REF model with the SH and SHLH experiments shows a reduction in CIN by around 35% outside of the green box and by 44% inside the green box. This reduction in CIN indicates that the atmosphere is less resistant to the initiation of convection, making it easier for fire-induced updrafts to develop. The CAPE values in the SH and SHLH experiments more than double outside the green box and increase by a factor of over 4.8 within the green box. Higher CAPE means that more energy is available for convection, which can lead to stronger and more vigorous updrafts. The impact of additional moisture release on CIN is less than 1%, and the increases in CAPE remain below 1.5%.

Figure 7 illustrates the temporal evolution of plume height for the cloud-free plume, the cloudy plume, and the plume area with clouds formed through pyro-convection. Figure 7a shows the cloud-free plume. It should be noted that at the beginning of the simulation, the plume is dense and concentrated close to the fires. This means that the number of grid points that

**Figure 7.** Temporal evolution of the plume top height throughout the day (AEDT) for three experiments: REF (black), SH (red), and SHLH (purple) for a) the plume area without clouds, b) the plume area where also clouds are present, and c) the area where a pyro-convective cloud has formed.

mask the plume is initially small but increases as the plume spreads. Therefore, during the first 8 hours of the simulation, the number of cloud-free grid points remains below 100. During the day, the number of masked grid points is on the order of 10000. The diagnosed plume top height remains around zero in the morning. After 12:00 AEDT, the top height increases for all experiments and decrease from the evening onward. The evolution of the top height in REF demonstrates the diurnal cycle of the atmosphere, which allows for enhanced vertical movement in the afternoon. The SH and SHLH experiments further emphasize the buoyancy created by the fire, which is also influenced by a diurnal cycle. Comparing the SH and SHLH experiments outlines the impact of the buoyancy from the sensible heat is dominant.

Figure 7b displays the evolution of the plume where clouds are also present. This mask includes many more grid points than the no-cloud mask, with grid point numbers on the order of  $10^5$ . In experiments accounting for fire-induced heat, there is an increase in plume top height of up to 500 m in the morning. A significant increase around 12.2 km is observed for all experiments around 13:00 AEDT, corresponding with the emergence of the first convective cloud cell, as the atmosphere becomes more unstable throughout the day and fire activity peaks. The impact of moisture release on the plume top height is small, with an average increase of 30 m. The REF experiment further shows a decrease in plume top height at 21:00 AEDT, which can be explained by the lower concentrations in the upper levels that no longer exceed the threshold and the transport of the elevated plume out of the domain.

Lastly, Figure 7c shows the evolution of the plume height within the newly forming pyro-convective cloud. Here, the masked grid points only exceed 100 in the afternoon, with masked grid points on the order of 10<sup>4</sup> for the SHLH and SH experiments, and on the order of 10<sup>3</sup> for the REF experiment. The plume top height is zero up to 12:00 AEDT, as no pyro-convective cloud has formed. Then, the height increases steeply for the experiments and decreases for REF and SH around 20:00 and for SHLH at 21:00 AEDT onward. The effect of heat and moisture release becomes evident when comparing the top heights during the formation of pyro-convective clouds. Fire-induced heat release increases the top height by up to 6.9 km. Further, the SHLH experiment indicates through a more uniform profile and later decline, that moisture release increases the pyro-convective cloud formation, and more grid cells exceed the threshold and for a longer time. We see that the enhancement of the plume top height is more visible in cloud-free and pyro-cloud regions. For regions with pre-existing clouds, additional buoyancy is most noticeable in the morning.

# 4 Discussion






First of all, we discuss the uncertainties, which contribute to the differences between the observations and the model. The retrieval itself is subject to uncertainties. The height retrievals depend on the relative viewing geometry of LEO-GEO. The retrieval process estimates the uncertainty of the retrieved parameters using a covariance matrix. This covariance matrix calculates the uncertainty statistics for, beyond other parameters, the retrieved height. The uncertainty derived from the covariance matrix serves as an effective guide to the quality of the retrievals (Carr et al., 2019). The shown retrievals are given with and error range of  $\pm$  200 to 300 m, which is consistent with the range reported by Carr et al. (2019).

Additionally, uncertainties persist in the comparison, as it is unclear at which specific mass mixing ratio or optical thickness

the plume or cloud is detected by the satellite. Therefore, the comparisons with simulations are strongly dependent on the threshold chosen for the plume and cloud definition. Furthermore, the vertical resolution of the model, which increases with height, introduces a varying uncertainty. The model's vertical resolution in the altitudes between 9 and 15 km ranges from 200 to 250 meters. This is comparable to the error range of the observations.







In contrast to the typical diurnal cycle of atmospheric stability and fire intensity, which suggests pyroCb clouds form in the early to late afternoon, some of the most intense pyroCb activity during the ANY event was observed at night (Peterson et al., 2021). This discrepancy between the diurnal cycle of atmospheric stability, fire intensity and the nighttime pyroCb activity is not captured in the simulation. We showed that the measured FRP strongly deviates from the assumed diurnal cycle function for our case study.

Another significant source of uncertainty is the input variables from GFAS, which are based on MODIS FRP measurements. These measurements are affected by interference from clouds and dense smoke plumes. Observations in Figure 2 show that the pyroCb clouds are close to the fire source. This suggests a possible underestimation of the fire intensity, leading to reduced aerosol, heat, and moisture emissions (Kaiser et al., 2012). Consequently, it becomes challenging to generate sufficient buoyancy to trigger pyro-convective cloud formation.

Further uncertainties are introduced by the analysis methods. The results depend on the selection of thresholds for the plumes and clouds, and therefore the plume heights. The discussed uncertainty due to the vertical resolution becomes critical when calculating the mean of the 100 largest altitudes. This introduces a dependency on the grid cell height, which increases with altitude. This dependency is problematic because it can lead to an overestimation of plume top heights in regions with coarser vertical resolution. As altitude increases, the grid cell heights become larger, and the mean value calculation may disproportionately represent higher altitudes, skewing the results. However, we use this approach because it provides a consistent method for identifying the highest plume altitudes across different scenarios. By focusing on the 100 largest altitudes, the method ensures that the most significant plume heights are captured, even if the vertical resolution varies. This consistency is crucial for comparative analysis and for understanding the overall behavior of plume dynamics in different atmospheric conditions.

Furthermore, our simulations do not include aerosol-cloud interactions. While studies such as Andreae et al. (2004); Koren et al. (2005); Wang et al. (2009) report enhanced updrafts due to these interactions, Luderer et al. (2006) found that although aerosol loading significantly alters the microphysical structure of pyro-convective clouds, the influence of cloud condensation nuclei on the dynamic evolution of the pyroCb remains limited. More recent studies by Kablick III et al. (2018) indicate that the impact of fire-generated aerosols on the development of a specific pyroCb were negligible compared to the effects of fire-generated heat fluxes. Therefore, we assume the effect of aerosol-cloud interaction is small.

Biomass burning aerosols have the ability to scatter and absorb solar radiation, which can play a significant role in plume rise, as shown by Ohneiser et al. (2023). Moreover, Chang et al. (2021) report aerosol-radiative forcing values between -14.8 and -17.7 W m<sup>-2</sup> close to the source, which is in the order of GFAS FRP for smaller fires. However, we focus on intense fires with pyro-convective cloud formation and therefore assume that in the mixture of aerosol-clouds, the neglect of the aerosol radiative effect is valid.

Further, the absence of condensation and gas-phase chemistry is a coarse simplification, but according to the findings of June

et al. (2022) condensation and gas-phase chemistry show minor impact on aerosol size distribution changes, whereas coagulation significantly contributes to particle growth in the early phase of plume development, which we account for.

Our analysis of the impact of additional water vapor agrees with the findings of Trentmann et al. (2006), who state that the emission of water vapor by the fire does not significantly contribute to the energy budget of the convection. However, our results show that moisture release by the fire increases the LWP and IWP, thereby enhancing latent heat release, especially in the center and eastern part of the domain. Therefore, the findings by Luderer et al. (2006), who observed that water vapor plays a less significant role in determining the injection height but enhances the amount of particles reaching upper levels further agrees with our results.

# 5 Conclusions







This study highlights the critical role of fire intensity and atmospheric stability in pyro-convective cloud formation and discusses remaining uncertainties through a direct comparison with observational data. These results demonstrate that the developed parameterization of heat and moisture release, based on satellite retrievals provided by GFAS, enables the simulation of pyro-convective clouds. However, the developed setup exhibits a temporal delay in the formation of these pyro-convective clouds when compared to observational data. These descepancies can be attributed to errors in the assumed diurnal cycle of fire activity. Sensitivity studies of the diurnal cycle function could address this issue and help reproduce the observations more accurately. By systematically varying the parameters of the diurnal cycle, we can better understand how different assumptions impact the formation of pyro-convective cells.

Overall, the simulations successfully captured the formation of pyro-convective clouds during the ANY event. The background meteorology, characterized by a highly unstable atmosphere, allows for the formation of convective cloud cells independently of the heat and moisture generated by the fire. These cells are partially fueled and intensified by the heat and moisture released. Additionally, in the center of the domain, a pyro-convective cloud forms only when accounting for sensible heat and moisture release. This simulation of a real case, involving several fire areas in close proximity, highlights how sensitive cloud formation and plume height are to fire intensity and background meteorology. The different fire areas exhibit different effects on cloud formation and plume height. Sensible heat release has been shown to be the predominant contributor to the formation of pyro-convective clouds. However, the release of moisture enhances cloud formation in the early stages of the formation process, which slightly increases the height of the plumes but increases the amount of aerosols lifted. Further case studies are needed to assess the general applicability of this parameterization in other regions. The developed method can potentially enable the simulation of pyro-convective clouds forming in close proximity to their actual occurrences anywhere on the globe.

Code availability. The ICON and ART models are openly available and accessible through the following link: https://icon-model.org/. The used code version is close to version 2024.10 (https://doi.org/10.35089/WDCC/IconRelease2024.10), certain code components that are

relevant for this work but not open-source conform can be made available upon reasonable request to the corresponding author. Access to the NASA 3D Wind Algorithm was granted by Dr. Jim Carr (jcarr@carrastro.com), and it is subject to his approval for access. Analysis and plotting tools were adapted form the following repositorty: https://github.com/alihoshy/art\_pytools.

Data availability. The model output from ICON-ART simulations generated in this study will be made available on Radar4KIT with an according DOI after the review process. Himawari-8 datasets are publicly accessible through Amazon Web Services (AWS). AWS description page: https://registry.opendata.aws/noaa-himawari. The MODIS datasets are also publicly accessible and downloaded from https://ladsweb.modaps.eosdis.nasa.gov/search/order/1/MODIS. The MODIS active fire product can be accessed here: https://earthdata.nasa.gov/earth-observation-data/near-real-time/firms/mcd14ml.

# Appendix A

**Figure A1.** Implemented diurnal cycle function of the emission over the day. The different colors represent the vegetation classes: tropical forest (brown), savanna (orange), and grassland (green). The vegetation class in the experiments is primarily tropical forest.

**Figure A2.** Plume and cloud top heights simulated for the REF experiment for December 30, 2019, at 10:30 AEDT (left) and for December 30, 2019, at 10:30 AEDT (right)

Author contributions. Lisa Janina Muth: Writing – review & editing, Writing – original draft, Visualization, Validation, Investigation, 385 Methodology, Software. Sascha Bierbauer: Writing – review & editing, Software, Validation. Corinna Hoose: Writing – review & editing, Supervision. Heike Vogel: Writing – review & editing, Software, Validation. Bernhard Vogel: Writing – review & editing, Conceptualization, Supervision. Gholam Ali Hoshyaripour: Writing – review & editing, Conceptualization, Supervision.

Competing interests. The authors declare that they have no competing interests.



Acknowledgements. This study contains modified Copernicus Atmosphere Monitoring Service information [2023]. Furthermore, resources of the Deutsches Klimarechenzentrum (DKRZ) granted by its Scientific Steering Committee (WLA) under project ID bb1070 have been used. This work contributes to and is partly funded by the project *PermaStrom* (grant no. 03EI4010A) within the seventh Energieforschungsprogramm of the German Federal Ministry of Economic Affairs and Climate Action (Bundesministerium für Wirtschaft und Klimaschutz, BMWK). L.M thanks the Graduate School for Climate and Environment (GRACE) for the support and opportunities that have contributed to this work. We acknowledge the use of the NASA 3D Wind Algorithm developed by Dr. James Carr and his team at NASA Goddard Space Flight Center, especially Dr. Mariel Friberg, with Dr. Dong Wu as the group leader and sponsor of this work. This algorithm has been instrumental in our analysis of plume and cloud height assignments. We appreciate the German Weather Service (DWD) for providing helpful analysis products for this research.

During the preparation of this work the authors used Microsoft Copilot in order to speed up the writing process and help with formulations. After using this tool, the authors reviewed and edited the content as needed and take full responsibility for the content of the published article.

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
