# Peer review of "Influence of Fire-Induced Heat and Moisture Release on Pyro-Convective Cloud Dynamics during the Australian New Year's Event: A Study Using Convection-Resolving Simulations and Satellite Data"

_EGUsphere, 2025_

## Author Response (AR1)

**Response to reviews**

We would like to express our gratitude to the referees for their thorough review and valuable feedback on our manuscript. We have carefully considered all the comments and suggestions provided. Below, we present a detailed point-by-point response to each comment, specifying the changes made in the revised manuscript. We believe these revisions have significantly improved the quality and clarity of our work. In the following the reviewers comments are bold, for better overview.

RC1: urlhttps://egusphere.copernicus.org/#RC1 'Comment on egusphere-2025-402', Anonymous Referee #1, 26 Mar 2025

The present manuscript investigates the formation of pyroconvection using the Australian New Year's event as an example. The results of sub-kilometer atmospheric simulations are described and compared with satellite data. For a high-frequency representation of fire emission fluxes, the GFAS data product, which is only available for daily mean values, is extended to include a description of a daily cycle. In sensitivity experiments, the effects of heat and water vapor fluxes are specifically investigated and compared with a reference case without corresponding fluxes. The study uses realistic atmospheric simulations to demonstrate the complexity of the formation of pyroconvection and emphasizes the special role of atmospheric stability for the convective transport of smoke aerosol into the upper troposphere.

The manuscript summarizes a carefully conducted scientific study. The methods used are appropriate for the research question and the presentation of the results is clear and logically structured. The figures are easy to understand and of high quality. I recommend publication in ACP, provided that the comments below have been considered and appropriate changes incorporated into the manuscript.

**Main comments:**

1. The present work proposes a scaling of the flux of sensible heat and water vapor emission from fires that has a pronounced maximum in the early afternoon. This scaling, which has been derived in the existing literature as typical behavior for pyroconvection, does not at all match the observed behavior during the ANY event, for which strong pyroconvective activity was documented during the nighttime hours. In other words, the present study suffers from the logical inconsistency that a generally valid parameterization for pyroconvective fire emissions is applied to an extreme case that deviates from the rule and does not follow this typical course. The authors address this aspect in their discussion section, but they need important parts of the presentation of the results to discuss deviations in the temporal course of the pyroconvection, which probably come about due to the insufficient assumptions in the parameterization. The general inconsistency will not be resolved with the existing material, but a preceding presentation of the temporal course of the Australian fire in comparison to the typical pyroconvective evolution may help to better understand the assumptions made and their influence.

R: We acknowledge that the typical behavior for pyroconvection, which shows a pronounced maximum in the early afternoon, does not align with the observed behavior during the ANY event, where strong pyroconvective activity was documented during nighttime hours. However, the primary motivation of our study was to develop a generally applicable setup for convection-resolving simulations of pyro-Cb events. This setup was tested using the Australian fire event as a case study. We aimed to create a parameterization that could be broadly applied to various pyroconvective scenarios, recognizing that each fire event has unique characteristics.

Further, we emphasize that every fire is unique, which inherently makes the development of a universally applicable parameterization challenging. The deviations observed in the temporal course of the pyroconvection during the ANY event highlight the complexities involved in modeling such phenomena. In the revised manuscript, we have expanded the discussion section to include a more detailed analysis

of the deviations in the temporal course of the Fire Radiative Power (FRP) observed and assumed in

the simulation during the ANY event. Additional material, including a new Figure (Figure 3), has been added to section 3.1.

"The fire input data remain a major source of uncertainty. The daily mean values provided by GFAS and the assumed generally applicable diurnal cycle are crude approximations. This is illustrated in Figure 3. Firstly, it is evident that the magnitude of the assumed FRP for the simulation underestimates the FRP observed by MODIS. This is expected, as the assumed FRP for the simulation is based on GFAS, which provides a daily mean value averaged over a  $1^{\circ} \times 1^{\circ}$  grid, compared to the MODIS active fire data at a 1 km resolution. The MODIS data have limited measurements due to the overpass times; however, they do not show a clear diurnal cycle as assumed in the parameterization. On the contrary, the measurements indicate high FRP during the night. As mentioned, for this particular event, high nighttime fire activity and pyroCb formation were reported."

While sensitivity studies of the diurnal cycle function would be valuable to better reproduce the observations, they are beyond the scope of the current work. However, we acknowledge the importance of such studies and suggest them as a direction for future research.

2. The presentation of the results is very descriptive and emphasizes unimportant details. I would have liked to see more focus on the convective processes involved and an exploration of the mechanisms for pyroconvective transport based on the case study. Furthermore, I would like to emphasize that the discussion section needs to be revised. Here, the limitations of the study and the resulting implications need to be addressed in more detail. An additional summary of the results without reflection is not needed here.

R: We revised the text according to this comment to make it more focused on the major outcomes of the study. We also expanded the description of the limitations in the discussion. Please, see the highlighted text in Sections 3 and 4 (1.220-351)

**Minor Comments:**

- 1.13 - 15: Please shorten and rephrase the sentence. Be clearer about what you want to communicate.

R: The sentence is rephrased to:

"Comparisons with observational data show that the plume's distribution and height are underestimated. However, the simulations align well with observations after a 5-6 hour delay, indicating that pyro-convective cells are accurately modeled but occur later than observed."

- 1.20: The Luderer reference is actually supporting the argument about diurnal changes.

R: The Luderer reference indeed supports the argument about diurnal changes. The reference analyzes the impact of a passing cold front, which highlights the strong diurnal changes referred to in line 20 throughout the day.

"However, there are exceptions to this typical diurnal cycles of the meteorology and fires. As outlined by Luderer et al. (2006), cold fronts can induce significant temperature drops before sunset, which are substantially greater than the usual diurnal variations experienced in the late afternoon."

- l. 26: Balch actually states for the Australian event that "Night-time MODIS active fire detections represented 76% [...] of total detections" – not supporting your argument here.

R: You are correct, this quote does not support our initial argument.

However, Balch also outlines exceptions and emphasizes the need for further studies due to the increasing deviations from the previously thought smaller nighttime fire activity. The primary focus of this part of the introduction is on general fire-atmosphere interactions, but we have included the previously made point that the ANY event is an exception to the general assumption of fire activity, and referenced it accordingly.

This is revised in the text as following:

"This leads to a majority of pyroCb clouds forming and reaching maturity in the late afternoons (Fromm

et al., 2010). ...

Furthermore, recent studies have shown an increase in nighttime fire activity, particularly in larger wild-fires. This increase is attributed to warmer and drier nighttime conditions, which can sustain fire activity throughout the night (Balch et al., 2022)."

- 1.27 (and 1.400) Fromm reference is wrong. I guess it should be the untold story paper

R: Corrected

- paragraph starting at l. 36: Here you jump from climate change, to interactive fire simulations to factors at impact pyroconvective transport. Please improve the line of argument here and use paragraphs as structural elements to separate between topics.

R: This section is rewritten.

"The effect of fire on meteorological variables and, consequently, on pyro-convective cloud formation is often not included in global and regional models. This omission leads to errors in the injection height of gases and particles, subsequently affecting their transport. To parameterize these processes accurately, a comprehensive understanding of the interplay between fire-induced buoyancy, latent heat release, and atmospheric stability is essential. Significant progress has been made in understanding these phenomena. Numerous studies with coupled fire-atmosphere models have addressed the uncertainties of fire-atmosphere interactions by accounting for fire dynamics. For example, research by Clark and Packham (1996); Clark et al. (2004) employs fine grid resolutions ranging from 4 meters to 120 meters ..."

- l. 60: "plume production": Please change wording.

R: Changed to:

"The smoke emission and dispersion are influenced by ..."

- l. 76: Start a new paragraph with "In this study..."

R: Done.

- l. 76-77: "intense fire-atmosphere interaction...": Please rephrase! This is misleading. You don't conduct studies on an interactively coupled fire-atmosphere system.

R: Changed to:

"In this study, convection-resolving simulations are performed and analyzed to determine if the atmospheric impact of these intense fires and to answer the following research questions: ..."

- Sect. 2.2: The description of aerosol emission fluxes is missing here. Do you also scale them with d? Please mention ICON and ART versions and refer, if possible, to a specific version tag.

R: As stated in the code availability statement, the ICON and ART code is generally open source. However, the code version used for this specific application contains parts that are not open source but are close to version 2024.10. You can find more information about this version at https://doi.org/10.35089/WDCC/IconRelease2024.10. This information has been added to the code availability statement. "The ICON and ART models are openly available and accessible through the following link: https://icon-model.org. The used code version is close to version 2024.10 (https://doi.org/10.35089/WDCC/IconRelease2024.10), certain code components that are relevant for this work but not open-source conform can be made available upon reasonable request to the corresponding author."

The aerosol emission is added. Please, see the highlighted part in section 2.2 (l.130-135). Yes, the aerosol emission is also scaled with the diurnal cycle function.

- Sect 2.3: Please add information on the domain size (horizontal extent) and typical vertical layer spacing to the description.

R: The horizontal extend and minimum and maximum layer thickness are added in the model configurations.

"The level thickness increases from in average 95 m in the lowermost level to 550 m at the top."

"For this work, a limited area mode simulation is performed. The area of the domain is shown in Figure 1 in the black box, which is approximately 340 km in length (north-south) and 230 km in width (east-west)."

- l. 128: "complex microphysical proceses" → Unclear what is meant here.

R: Dipankar et al. (2015) refer to the double-moment microphysics scheme by Seifert and Beheng (2006), which can be enabled for simulations at this spatial resolution. However, since a single-moment scheme is used in this study, this part is removed from the text.

- end of Sect 2.3: Please add the values of peak sensible heat and peak fire-induced water vapor flux to the description. Please also provide the equivalent latent heating potentially released by condensing the additional water vapor.

R: Thank you for pointing out these important details, they have been added to section 2.3.

"During the simulation, a peak sensible heat release of  $2.24~\rm kW~m^{-2}$  is reached. The peak fire-induced water vapor flux reaches  $1.07\times10^{-6}~\rm kg~m^{-2}~s^{-1}$  and assuming the latent heat of condensation for water is approximately  $2257~\rm kJ~kg^{-1}$ , the a potential additional latent heat flux can reach a maximum of  $0.02~\rm W~m^{-2}$ ."

- l. 194: "Besides ..." This is too detailed and insignificant. Please delete it!

R: Done.

- l. 202: "However ..." Again irrelevant information here.

R: Done.

- Table 1: Please use positive CIN value following the standard definitions e.g. of AMS glossary. Please only print significant digits in the table! Please support your simulated values with observations. Please provide a comparison of your simulated profiles to radiosonde data in your response.

R: We now use positive CIN values and removed the insignificant digits. The comparison of simulated values with observations would indeed be helpful for validating our results. However, we have not found any radiosonde data or other meteorological observations in the domain and time of our simulations, making it challenging to make a robust comparison.

Despite this limitation, we have compared our simulated profiles of REF and SHLH with radiosonde data from Wagga Wagga (shown in R2 and R3). The location of Wagga Wagga is marked with the magenta star in R1, which is just outside the domain.

The radiosonde data from Wagga Wagga on December 30 at 00:00 UTC is compared to the mean profiles in the fire area of the green box, where the fire has the strongest effect. Due to the location mismatch, no solid conclusions can be derived from the comparison with the soundings.

l. 220: CAPE: Please clarify which CAPE definition is used – surface-based, most-unstable, mixed-layer, etc...

R: The CAPE definition used in this context is surface-based CAPE. This means that the calculations are based on the conditions at the surface level.

"Table 1 shows how the fire-atmosphere interaction impacts surface-based Convective Inhibition (CIN)

R 1: Simulation domain outlined in the blue box, area of most intense fire in green box, Wagga Wagga, location of the radiosonde launch marked with the magenta star.

R 2: Radiosonde Sounding at Wagga Wagga on December 30 at 00:00 UTC of Temperature: REF experiment in red, sounding in brown, Dew Point Temperature: REF experiment in dark green, sounding in lime

R 3: Radiosonde Sounding at Wagga Wagga on December 30 at 00:00 UTC of Temperature: SHLH experiment in red, sounding in brown, Dew Point Temperature: SHLH experiment in dark green, sounding in lime

and surface-based Convective Available Potential Energy (CAPE) of the green box (GB) and elsewhere for December 30, 16:00 AEDT. The average CIN and CAPE values are calculated within the fire area for the corresponding experiments."

**- l. 234-236: Plume definition should be moved to method section. The 100 largest altitudes definition introduces dependence on the grid which is sub-optimal.**

R: Yes, calculating the mean value of the 100 largest altitudes introduces a dependency on the grid cell height, which increases with altitude. This dependency is problematic because it can lead to an overestimation of plume top heights in regions with coarser vertical resolution. As altitude increases, the grid cells become larger, and the mean value calculation may disproportionately represent higher altitudes, skewing the results.

However, this approach is justified because it provides a consistent method for identifying the highest plume altitudes across different scenarios. By focusing on the 100 largest altitudes, the method ensures that the most significant plume heights are captured, even if the vertical resolution varies. This consistency is crucial for comparative analysis and for understanding the overall behavior of plume dynamics in different atmospheric conditions. We clarify this uncertainty now in the Discussion section.

**- l. 235: "the sum of LWC+IWC [...] must be larger..." This is not clear! Do you sum up values of an intensive quantity?**

R: There was an error in the initial statement. The threshold should indeed be smaller, as it is used to allow for background concentration. This is corrected and part is moved to Section 2.5.

"The plume is considered cloud-free if the grid cell has a mass mixing ratio of LWC+IWC smaller than  $1\times 10^{-32}~{\rm kg~m^{-3}}$ . In the plume with clouds case, LWC+IWC within a grid cell must be greater  $1\times 10^{-32}~{\rm kg~m^{-3}}$ . The plume with pyro-convective clouds is defined by an increase in aerosol mass mixing ratio by  $1\times 10^{-6}~{\rm kg/m^3}$  and an increase of LWC+IWC by  $5\times 10^{-8}~{\rm kg~m^{-3}}$  within a grid cell compared to the REF experiment. "

**- paragraph starting with l. 257: There are too many methodological descriptions mixed in with the presentation of results. It is especially unclear how pyroCb is defined for REF (l. 260).**

R: We have moved the methodological descriptions to the methodology section (2.5), as suggested, to improve clarity in the presentation of results.

Regarding the REF experiment and the definition of pyroCb (line 260), since there is no pyro cloud in the REF experiment, the grid cells corresponding to the SHLH experiments are used. This clarification has been added to the methodology section.

"The presence of clouds within the plume is defined via a threshold for the sum of LWC and IWC within a grid cell. The plume is considered cloud-free if a grid cell has a mass mixing ratio of the sum of LWC and IWC smaller than  $1\times10^{-32}$  kg m-3."

**- l. 265: How do you define "lifespan"?**

R: The term "lifespan" refers to the amount of time during which a pyro cloud forms and exceeds the defined thresholds. This definition has been clarified in the relevant sections to ensure a better understanding. See 1.293.

"The effect of heat and moisture release becomes evident when comparing the top heights during the formation of pyro-convective clouds."

**- l. 270: "Our analysis shows ..." This is not clear! How and where do you show observational uncertainties? Please be more precise!**

R: The retrieval algorithm includes a calculation of the error, which is shown in the following figures R4 and R5. However, this analysis is not included in the paper as we do not believe it is of significant importance to the overall findings. We clarified the description in the text to ensure better precision and understanding.

"The retrieval itself is subject to uncertainties. The height retrievals depend on the relative viewing geometry of LEO-GEO. The retrieval process estimates the uncertainty of the retrieved parameters using a covariance matrix. This covariance matrix calculates the uncertainty statistics for, beyond other pa-

R 4: Clear-sky ground retrievals from the MODIS-AHI 3D-winds (December 30, 2019, at 10:30 AEDT) show the accuracies of retrieved heights (a) and velocities (c,d). The mean  $\mu$  and standard deviation  $\sigma$  over sample size N are computed for each histogram and the regression parameters (b) for retrieved height versus terrain height are reported for the ground-point class

R 5: Clear-sky ground retrievals from the MODIS-AHI 3D-winds (December 30, 2019, at 10:30 AEDT) show the accuracies of retrieved heights (a) and velocities (c,d). The mean  $\mu$  and standard deviation  $\sigma$  over sample size N are computed for each histogram and the regression parameters (b) for retrieved height versus terrain height are reported for the ground-point class

rameters, the retrieved height. The uncertainty derived from the covariance matrix serves as an effective guide to the quality of the retrievals Carr et al. (2019). The shown retrievals are given with and error range of  $\pm$  200 to 300 m, which is consistent with the range reported by Carr et al. (2019)."

- l. 293 - 307: The paragraphs appear to be a summary of the results presented before. To my opinion, this does not fit into the discussion section where the focus should be on limitations, implications and relation to existing knowledge.

R: This has been adapted. Please, view the highlighted changes in the discussion section.

- Open science: I recommend making analysis and plotting scripts openly available to improve the reproducibility of your results.

R: The following repository contains the python scripts to analyze and plot ICON-ART data, this is now cited in the manuscript under **Code availability**. The skips can be adopted and used based on the information provided in the text to reproduce the plots.

https://github.com/alihoshy/art\_pytools

RC2: https://egusphere.copernicus.org/\#RC2'Comment on egusphere-2025-402', Anonymous Referee #2, 19 Apr 2025

The present manuscript focusing on the 2019/2020 Australian New Year's bushfire event. This study analyzes the role of wildfire-induced heat and moisture fluxes in the genesis and development of pyroconvective clouds. The goal is to refine the understanding and numerical representation of these processes using the ICON-ART model for simulations, which includes fire effects like sensible heat and moisture release, and fire data obtained from the Global Fire Assimilation System (GFAS), based on MODIS satellite observations.

Three simulation setups were tested, REF, SH and SHLH. In addition, NASA's 3D Wind Algorithm was used to retrieve plume and cloud heights from satellite data.

The introduction is well written and effectively contextualizes the challenges involved in simulating pyroconvective clouds, highlighting the complexity of this type of atmospheric phenomenon.

In the methods and materials section, the model, its parameterizations, and configurations are clearly and objectively described, allowing for a good understanding of the adopted approach.

The presentation of the results is quite detailed. However, the excessive use of acronyms compromises the flow of reading, as the reader must constantly refer to the legend or glossary to understand the terms. This can be particularly challenging for readers who are not fully familiar with the field.

R: Thank you for your feedback. We have reduced the use of acronyms and ensured that terms are spelled out where necessary. Additionally, definitions and detailed descriptions of analysis methods have been relocated to the methodology section. These adjustments should enhance the flow of reading and make the main text more accessible, especially for readers who may not be fully familiar with the field. We believe these changes will significantly improve the overall readability and comprehension of the document.

In the discussion section, the authors appropriately address the uncertainty factors that may have contributed to the differences between the model results and the observational data. However, the way these uncertainties were estimated or calculated is not sufficiently clear. For example, in line 270, the authors state: "Our analysis shows that the height retrievals have an error range of  $\pm 200$  to 300 m," but it is not specified how this error range was obtained — it would be important to clarify whether this estimate was derived from comparisons with observational data, statistical analysis, or another method.

R: This has been pointed out by RC1, and changed to:

"The retrieval itself is subject to uncertainties. The height retrievals depend on the relative viewing geometry of LEO-GEO. The retrieval process estimates the uncertainty of the retrieved parameters using a covariance matrix. This covariance matrix calculates the uncertainty statistics for, beyond other parameters, the retrieved height. The uncertainty derived from the covariance matrix serves as an effective guide to the quality of the retrievals Carr et al. (2019). The shown retrievals are given with and

error range of  $\pm$  200 to 300 m, which is consistent with the range reported by Carr et al. (2019)."

Additionally, in line 129, the authors state: "However, this study does not consider aerosol-cloud and aerosol-radiation interaction." It would be relevant to further discuss what types of uncertainties this omission may introduce into the model's performance, especially considering that such interactions can significantly impact the development and evolution of convective clouds.

R: This is now addressed in the discussion.

"Furthermore, our simulations do not include aerosol-cloud interactions. While studies such as Andreae et al. (2004); Koren et al. (2005); Wang et al. (2009) report enhanced updrafts due to these interactions, Luderer et al. (2006) found that although aerosol loading significantly alters the microphysical structure of pyro-convective clouds, the influence of cloud condensation nuclei on the dynamic evolution of the pyroCb remains limited. More recent studies by Kablick et al. (2018) indicate that the impact of fire-generated aerosols on the development of a specific pyroCb were negligible compared to the effects of fire-generated heat fluxes. Therefore, we assume the effect of aerosol-cloud interaction is small

Biomass burning aerosols have the ability to scatter and absorb solar radiation, which can play a significant role in plume rise, as shown by Ohneiser et al. (2023). Moreover, Chang et al. (2021) report aerosol-radiative forcing values between 14.8 and 17.7 W m $^{-2}$  close to the source, which is in the order of GFAS FRP for smaller fires. However, we focus on intense fires with pyro-convective cloud formation and therefore assume that in the mixture of aerosol-clouds, the neglect of the aerosol radiative effect is valid."

I also suggest including a more in-depth discussion on the possible uncertainties associated with the plume height values obtained from the NASA 3D wind retrieval algorithm, as well as a comparison between the vertical resolution of these height estimates and the vertical resolution of the model used in this study. Are they consistent? Such an analysis would help better assess the compatibility between the data and the model results.

R: Thank you for your suggestion. We have included a more in-depth discussion on the possible uncertainties associated with the plume height values obtained from the NASA 3D wind retrieval algorithm. The retrieval algorithm has an estimated accuracy of 200-300 meters in height.

Regarding the comparison between the vertical resolution of these height estimates and the vertical resolution of the model used in this study, the model's vertical resolution in the altitudes between 9 and 15 km ranges between 200 and 250 meters. This is comparable with the error range of the observations.

"Additionally, uncertainties persist in the comparison, as it is unclear at which specific mass mixing ratio or optical thickness the plume or cloud is detected by the satellite. Therefore, the comparisons with simulations are strongly dependent on the threshold chosen for the plume and cloud definition. Furthermore, the vertical resolution of the model, which increases with height, introduces a varying uncertainty. The model's vertical resolution in the altitudes between 9 and 15 km ranges from 200 to 250 meters. This is comparable to the error range of the observations."

Would it be possible for the authors to consider incorporating data from other satellites to reduce the uncertainties related to the plume height estimates obtained by the model? For example, data from active remote sensing instruments?

R: Thank you for your suggestion regarding the incorporation of data from other satellites to reduce uncertainties related to plume height estimates. Due to the small simulation domain and limited simulation time, observations are indeed rare. Specifically, MISR and Cloudsat/CALIPSO missed the simulated domain during the study period. Observations from MODIS, such as cloud top height, are comparable to the height retrievals from NASA 3D wind shown in the manuscript, that incorporate MODIS data.

Overall, the article makes several important scientific contributions to the understanding and modeling of pyro-convective cloud dynamics, especially in the context of extreme wildfire events.

The study introduces the parameterization scheme for fire-induced sensible heat and moisture release using satellite data (GFAS).

The manuscript provides clear evidence that sensible heat is the dominant factor in driving plume rise and cloud formation, and demonstrates how wildfires can inject aerosols and trace gases into the upper troposphere and lower stratosphere, potentially affecting global atmospheric composition and radiation balance. By improving our ability to simulate how wildfires impact the atmosphere, the study helps bridge gaps between fire science, meteorology, and climate science. In this regard, I recommend the publication of the manuscript, provided that the reviewers' suggestions are adequately addressed.